# Dietary Curcumin Modulating Effect on Performance, Antioxidant Status, and Immune-Related Response of Broiler Chickens Exposed to Imidacloprid Insecticide

**DOI:** 10.3390/ani13233650

**Published:** 2023-11-25

**Authors:** Naglaa Z. Eleiwa, Ahmed A. El-Shabrawi, Doaa Ibrahim, Abdelwahab A. Abdelwarith, Elsayed M. Younis, Simon J. Davies, Mohamed M. M. Metwally, Ehsan H. Abu-Zeid

**Affiliations:** 1Department of Pharmacology, Faculty of Veterinary Medicine, Zagazig University, Zagazig 44519, Egypt; drnaglaa876@gmail.com (N.Z.E.); ahmedshabrawi@gmail.com (A.A.E.-S.); 2Department of Nutrition and Clinical Nutrition, Faculty of Veterinary Medicine, Zagazig University, Zagazig 44519, Egypt; 3Department of Zoology, College of Science, King Saud University, P.O. Box 2455, Riyadh 11451, Saudi Arabia; awarith@ksu.edu.sa (A.A.A.); emyounis@ksu.edu.sa (E.M.Y.); 4Aquaculture Nutrition Research Unit ANRU, Carna Research Station, Ryan Institute, College of Science and Engineering, University of Galway, H91V8Y1 Galway, Ireland; sjdplymouth@live.co.uk; 5Department of Pathology and Clinical Pathology, Faculty of Veterinary Medicine, King Salman International University, Ras Sudr 46612, Egypt; metywally@gmail.com; 6Department of Pathology, Faculty of Veterinary Medicine, Zagazig University, Zagazig 44519, Egypt; 7Department of Forensic Medicine and Toxicology, Faculty of Veterinary Medicine, Zagazig University, Zagazig 44519, Egypt

**Keywords:** curcumin, imidacloprid, immune-toxicity, hematological indices, phagocytosis, immune-related genes mRNA expression, oxidative stress, growth performance, 308 Ross Broiler

## Abstract

**Simple Summary:**

This experiment was conducted to examine the protective role of curcumin (CUR) in mitigating imidacloprid (IMI)-induced toxic effects on broilers’ growth performance, immune response, antioxidant status, and the expression of immune-associated genes. CUR significantly improved overall growth performance-related indices when compared to IMI-exposed birds. Additionally, CUR enhanced hematological indices, phagocytosis, and total protein, albumin, globulin, AST, ALT, lysozyme, and IgG levels when compared to the IMI-exposed birds. CUR supplementation significantly modulated oxidative stress-related indices, including TAC, SOD, CAT, and GPx, and decreased MDA levels when compared to IMI-exposed birds. The inflammatory response was modulated after CUR supplementation as supported by down-regulation of *IL-1β*, *TNF-α*, and *TLR4* mRNA expression levels and up-regulated the *IL-10* mRNA expression levels in the spleen when compared to IMI-exposed birds. These results collectively showed, for the first time, that CUR had an in vivo protective effect against IMI toxicity in broiler chickens.

**Abstract:**

Birds appear to be especially vulnerable to adverse impacts from insecticides. This is especially true for imidacloprid (IMI), which is considered the most toxic to avian species. Recently, prospective studies aimed at including natural alternative products to alleviate the toxic impact that comes from insecticides have been increased. Focusing on herbal growth promoters and antioxidative medicament for the poultry industry, this ongoing experiment was conducted to examine the curcumin role (CUR) in mitigating IMI-prompted detrimental effects on broilers’ performance, immunity, and antioxidant status. A total number of one hundred and fifty commercial meat-type Ross 308 broilers chicks (one-day-old) were randomly allocated into equal five groups (30 chicks/group and 10 birds/replicate). The first group (C) was the control; the second group (CUR) was fed a diet containing CUR at the level of 450 mg/kg; the third group (IMI) was fed control diet for 14 days and then was fed a diet containing IMI at the level of 50 mg/kg; the fourth group (CUR+IMI co-treated) was fed a diet containing CUR+IMI; and the fifth group (CUR+IMI pro/co-treated) was fed a diet containing CUR for 14 days as protective and then a diet containing CUR+IMI for the rest of the trial. CUR supplementation either in the (CUR pro/co-treated) or (CUR co-treated) groups significantly (*p* < 0.05) improved final body weight and total body weight gain while decreasing the total feed intake and feed conversion ratio when compared to the IMI-exposed and non-treated birds. CUR induced a significant (*p* < 0.05) enhancement in hematological indices, phagocytosis %, phagocytic index, intracellular killing capacity, total proteins, globulin, liver function enzymes, lysozyme activity, and immunoglobulin-G levels compared to IMI-exposed and non-treated birds. In addition, dietary supplementation of CUR significantly (*p* < 0.05) modulated oxidative stress-related biomarkers in splenic tissues (total antioxidant capacity, superoxide dismutase, catalase, and glutathione peroxidase) and decreased malondialdehyde levels (*p* < 0.05) when compared to IMI-exposed and non-treated birds. CUR significantly down-regulated mRNA levels expression of *IL-1β*, *TNF-α*, and *TLR4* and up-regulated *IL-10* mRNA expression levels in spleens of birds when compared to those exposed to IMI-and non-treated. Finally, our results provided new insight into IMI-induced immuno-toxicity in broiler chickens. Furthermore, for the first time, our study informed that CUR can cause an in vivo protective effect against IMI toxicity, principally as a protective and/or as concurrent supplementation during the exposure to IMI toxicity.

## 1. Introduction

Worldwide, poultry production has actively selected features that maximize birds’ output as well as boost birds’ immunity, thereby enhancing birds’ productivity [1], feed profitability and utilization [2], and their performance [3,4]. In such a scenario, the poultry industry sector has seen a rise in the valuation of plant-based products and their bioactive components that are targeted to improve birds’ health, productivity, and the products quality derived from them [5]. For broiler producers, boosting birds’ immunity is a critical area [6,7]. Various approaches have been used, such as adding feed additives like natural extracts [8] with antioxidant potentials [9,10,11,12] and to modulate the metabolic homeostasis and immune status of birds [13]. To this end, turmeric, also known as *Curcuma longa* Linn (member of the *Zingiberaceae* family), is one of the most widely used nutritional supplements worldwide [14,15]. Turmeric is a potential medicinal herb used as a natural feed supplement in chickens’ diets and has been used to replace dietary antibiotics with a beneficial impact on health [16]. According to Ashraf [17], CUR makes up the majority (80%) of the total amount of curcuminoids in turmeric powder. The hydrophobic phenolic compound gives turmeric its orange-yellow pigment [18]. It is mostly used to enhance food palatability, improve food appearance, preserve food, regulate aflatoxin-induced mutagenicity, and protect against hepatic carcinogenicity [19]. Owing to its capacity to affect a variety of signaling molecules, including inflammatory molecules, cell survival proteins, and drug resistance proteins, CUR has been described as a multi-target drug [20].

Abou-Elkhair, et al. [21], reported that CUR can be used as a feed additive in poultry to improve their growth performance. The free radical scavenging, anti-inflammatory activities, and immune response boosting are just a few of the therapeutic and pharmacological effects of dietary CUR [22,23,24]. According to the above-mentioned biological characteristics, dietary CUR has been identified as a possible feed additive to lessen the harmful effects of chronic heat stress in poultry [25]. Various experimental and clinical research data have reported that CUR is pharmacologically considered to be a harmless substance that possesses antioxidant, anti-inflammatory, and antimicrobial consequences [15,26,27].

Neonicotinoids are a group of globally used insecticides with systemic properties that were increasingly utilized from early 2000 onwards and are approved in 120 countries [28]. Besides, imidacloprid (IMI) has been established as a novel insecticide due to its molecular properties and distinct effect on pests [29]. Undesirable impacts on non-target species resulted from a wide spectrum of neonicotinoids [30]. Additionally, exposure to neonicotinoids has been linked to sub-lethal effects in birds, including neurobehavioral alterations, immunotoxicity, poor growth and development, and decreased reproductive efficiency [31,32]. Cestonaro, et al. [33], reported that the immune system could be challenged by neonicotinoids. Due to the association with decreased fitness and survival probability in birds, immunotoxicity is one of their notable sub-lethal impacts [34]. Neonicotinoids pesticides target nicotinic-acetylcholine receptors, which are found in several immune system components [35,36]. This may contribute to the immune suppression seen in bees, rodents, and birds [37,38,39].

Birds appear to be particularly susceptible to adverse impacts from neonicotinoids, together with IMI being the most utilized pesticide worldwide [40] and the most toxic insecticide to avian varieties [32,41]. Exposure to IMI through water or food occurs frequently as broiler chickens are the most common source of animal protein and are frequently raised using traditional farming practices, particularly in developing nations [42]. IMI is regularly found in food samples and drinking water samples at levels as high as 4.49 ng/g and 8.622 ng/L, respectively; this is due to its high water solubility and extended persistence [43,44]. IMI is applied to control insects that attack poultry farms and damage crops [45]. To combat the intense presence of insects, IMI spray is applied for utensil cleaning and wall painting [46]. Although IMI appears to be safe for birds to consume and selectively kills insects, it can still be harmful to birds because the safe dosage levels are unknown, and the chance of its ingesting through feed and water are high [47,48]. The breakdown products of IMI can also potentially cause immunosuppression directly or indirectly through triggering stress responses and the neuroendocrine system [49].

Dietary supplements containing natural products with antioxidant activities offer excellent possibilities for mitigating neonicotinoid toxicity since mitochondrial oxidative stress (OS) is a significant factor in neonicotinoid-induced toxicity [50]. Free radicals implicated in lipid peroxidation and OS are produced in large quantities as a result of IMI toxicity [51]. Osman et al. [52], found that birds exposed to IMI regularly exhibited disturbance of their antioxidative status, inflammation, and immunosuppression. Moreover, levels of liver-related indices (alkaline phosphate and transaminase), lipids (cholesterol and triglycerides), and hematological indices were altered in these birds [53]. Previous studies stated that feed supplements with vitamin E, selenium, and silymarin were found to reduce the harmful effects of IMI exposure in Japanese quails and chickens [46,54,55]. In layers, the hepatic tissue levels of GSH were dropped while TBARS were increased following exposure to IMI [56]. In addition to teratogenic concerns in white leghorn chicken [57], IMI exposure has an impact on embryonic development and chicks’ survivability in bobwhite quails [58]. Following IMI exposure, OS and lipid peroxidation were prompted in the RBCs, liver, kidney, and testes of white leghorn cockerels [59]. Therefore, it is advised to use natural antioxidants to restore the antioxidative ability, lessen lipid peroxidation, and control the physiological and metabolic processes inside the birds’ body [51,53]. In light of the aforementioned problems, the goal of the current study was to assess the effectiveness of pro and/or concurrent dietary additions of CUR to the diets of Ross 308 broiler chickens that had been exposed to IMI by evaluating the chickens’ growth performance, hematological indices, phagocytosis, immune and oxidative status, and mRNA expression profiling of immune-related genes.

## 2. Materials and Methods

### 2.1. Test Compounds and Chemical Reagents

Imidacloprid (IMI) (CAS No. 138261-41-3), analytical standard, purity grade (100%) PESTANAL^®^, Curcumin (CUR) (CAS: 458-37-7) with purity determined by HPLC to be more than 98% were obtained from Sigma-Aldrich Laborchemikalien GmbH; Germany. The total *RNA* extraction reagent was Trizol™ (Invitrogen; Thermo Fisher Scientific, Inc., Waltham, MA, USA), and the HiSenScript™ RH (-) cDNA Synthesis Kit was also used (iNtRON Biotechnology Co., Seongnam, Kyonggi-do, Republic of Korea).

### 2.2. Experimental Birds, Design, and Diet Preparation

The protocol of the study complied with the principles and regulations approved by Zagazig University’s Faculty of Veterinary Medicine’s Institutional Animal Care and Use Committee (IACUC) (ZU-IACUC/2/f/75/2022).

One hundred and fifty chicks (one-day-old) of a meat type (Ross 308) with an average body weight of 45 ± 5 g and provided by a commercial chicks’ producer (Dakahlia Poultry, Mansoura, Egypt) were used for the current study. On arrival, they were weighed then equally divided randomly into five experimental groups. Each group contained 30 chicks (three replicates/treatment, and 10 birds/replicate). Study groups included the first control group (C), fed a normal control diet; the second group (CUR), the curcumin group, fed a supplemented diet with 450 mg/kg of CUR, according to Cheng et al. [60]; the third imidacloprid group (IMI), fed on control diet for 14 days then fed on a supplemented diet with a dose of IMI 50 mg/kg diet following Ravikanth et al. [61]; the fourth group, CUR+IMI co-treated, fed the control diet for 14 days then fed a supplemented diet with CUR+IMI until the end of the experiment; the fifth group, CUR+IMI pro/co-treated, curcumin + imidacloprid protected/co-treated group fed a supplemented diet with CUR for 14 days then fed a supplemented diet with CUR+IMI until end of the experiment (4 weeks). Experimental groups and the treatment design are shown in Figure 1. Birds were raised in an open home with natural ventilation in the Faculty of Veterinary Medicine at Zagazig University’s animal research unit. Birds were reared in batteries with an automatic water system, ban feed in front of birds, and at a density of 10 birds/m^2^. Lighting regime was 24 h from days 1 to 3 and then 23 h lighting was applied up to the end of the experiment. The starting temperature was adjusted to 33 ± 1 °C for the first 3 days and then decreased by 3 °C each week until it reached 24 °C at the end of the experimental period according to the Aviagen guidelines [62]. Humidity was maintained at around 60% throughout the whole experiment.

The basal isocaloric and isonitrogenous diets were formulated following Ross Broiler-Pocket Guide specifications from Aviagen [63]. The chemical composition and nutrient concentrations of the basal diets are listed in Table 1. The experiment lasted for seven weeks. Diets for the starter stage (1–10 days), grower stage (11–20 days), and finisher stage (21–42 days) were available throughout the experiment. The utilized feedstuffs and the experimental diets’ proximate chemical analyses (for moisture, crude protein, and ether extracts) were conducted following AOAC [64] and are shown in Table 2. Birds received vaccinations against Newcastle disease at 7 and 14 days of age and against Gumboro disease at 11 and 22 days of age, according to Giambrone et al. [65]. Birds of all groups were monitored for any disease challenge or mortalities over the experimental period.

### 2.3. Total Growth Performance Related Indices

To obtain the average initial body weight (IBW), each bird was weighed separately on the first day of life. The body weight was then calculated each week by dividing the total bird weights by the number of birds in each group. According to procedures outlined by Kishawy, et al. [68] and Ibrahim, et al. [69], the total body weight gain (TBWG), total feed intake (TFI), total feed conversion ratio (TFCR), and final body weight (FBW) per each replicate for the entire experimental period were estimated.

### 2.4. Collection of Samples

Six birds per group were randomly chosen at the end of the 42-days feeding study, fasted overnight, and weighed. At the end of the experimental period, blood from each group was aseptically collected from the wing vein and separated as follows: one part was collected into clean and dry 15 mL falcon tubes without the use of an anticoagulant, allowed to clot at room temperature, and then centrifuged for five minutes at 3000 rpm for serum separation and kept at −20 °C until biochemically analyzed for protein and lipid profiles, liver functions, and immunological markers. Another part of blood was collected in sterile tubes containing dipotassium Ethylene di amine tetra acetic acid salt (EDTA) for hematological examination. The third part of blood was collected in heparinized tubes for phagocytosis assay.

Splenic tissues (1 g) were collected, cleaned with an ice-cold 0.85% NaCl solution, and homogenized in 9 mL of ice-cold phosphate buffer saline (PBS, pH 7.5). For the purpose of estimating oxidative stress-related indices, the homogenate was spun in a cooled centrifuge for 15 min at 3000 rpm. The supernatant was then collected into Eppendorf tubes and maintained at −80 °C.

Small parts of splenic tissues were dissected for molecular analysis immediately after scarification and maintained in TRIzol reagent at −80 °C until further examination by the RT-qPCR assay of immune-related genes including Toll-Like Receptor 4 (*TLR-4*), Interleukin 1 Beta (*IL-1β*), and Interleukin 10 (*IL-10*). Other splenic tissues were stored for histopathological and immunohistochemical analysis in neutral buffered formalin (10%) at room temperature.

### 2.5. Hematological Analysis and Phagocytosis Assay

Haematological parameters were measured using Hemascreen 18 Automatic Cell Counter (Hospitex Diagnostics, Sesto Fiorentino, Italy), on the basis of Harrison and Harrison [70], including red blood cells (RBCs), hemoglobin (Hb), hematocrit value (Hct), total leukocyte counts (WBCs), and differential leukocyte counts were determined as described by Nengsih and Mustika [71].

The phagocytic assay was carried out as described by Bos and de Souza [72], with some adjustments. Blood was collected quickly followed by the extraction of a peripheral blood mononuclear cell layer, washing, re-suspension in Roswell Park Memorial Institute (RPMI-164) medium, and the addition of 15% fetal calf serum (FCS). Then, 5 × 10^6^ mononuclear cells were seeded in a 1 mL volume for culture in chambers with coverslips; these were stained and incubated for 1 h at 37 °C with 5% CO_2_ and 99% humidity to create a monolayer of macrophages. After washing three times to eliminate non-adherent cells, adherent macrophages were treated for 24 h with 1 mL of *Candida albicans* (10^7^/mL of RPMI with 15% FCS) before being washed three times, fixed, and stained. In order to calculate the percentage of phagocytic macrophages (number of phagocytic macrophages/total number of macrophages), 100 macrophages were counted.

### 2.6. Biochemical Measurements of the Serum and Splenic Tissue

The serum lysozyme activities of broilers were determined according to Ellis [73]. Serum IgG levels were determined with chicken immunoglobulin (IgG) *ELISA* kits obtained from MyBioSource with Catalog number: MBS260043.

Colorimetric diagnostic kits from Sigma Aldrich were used for estimation of serum total cholesterol (TC), triglycerides (TG), high-density lipoprotein-cholesterol (HDL-C), low-density lipoprotein-cholesterol (LDL-C), and very low-density lipoprotein (VLDL) levels, with product number MAK043 for TC, TR0100 for TGs, and MAK045 for HDL and LDL/VLDL assay Kits, following the manufacturer’s instructions. The serum levels of albumin and total protein (TP) were determined by ready-made diagnostic kits provided by Agappe diagnostics kits, Code No: 51001002. The serum globulin (Gl.) levels were calculated mathematically by subtracting albumin values from total proteins as described by Doumas, et al. [74]. Serum levels of aspartate aminotransferase (AST) and alanine aminotransferase (ALT) were determined by ready-made diagnostic kits provided by Agappe diagnostics kits, Product. NO.: 51408002 and 51409002, respectively.

Splenic levels of total antioxidant capacity (TAC), catalase (CAT) activity, Superoxide dismutase (SOD) activity, and Glutathione peroxidase (GPx) was determined by colorimetric determination method by ready-made diagnostic kits provided by Bio-diagnostic, Egypt CAT. NO. TA 25 13, CA 25 17, SD 25 21, GP 2524, respectively. Malondialdehyde (MDA) levels were determined by enzymatic colorimetric method by using ready diagnostic kits provided by Bio-diagnostic, Egypt, CAT. No. MD 25 29.

### 2.7. Transcriptional Analysis of Immune-Related Genes IL-1β, TLR-4, and IL-10 in Spleen Tissue Using Quantitative Real-Time PCR

Splenic tissue samples were used for detecting the mRNA expression levels of immune-related genes. Extraction of total RNA was completed via the QIAamp RNeasy Mini kit (Cat. No. 51304; Qiagen, Hilden, Germany) following the instructions of the manufacturer. The extracted RNA concentration was detected at 260 nm and the clarity of RNA was detected by Spectrostar NanoDrop TM 2000 spectrophotometer (Cat. No. ND-2000; Thermo Fisher, Santa Clara, CA, USA). The Stratagene MX3005P real-time PCR machine (Cat. No. PF1457N; Thermo Fisher, Santa Clara, CA, USA) was utilized for one-step RT-qPCR amplification, in triplicate, via a Quanti Tect SYBR Green RT-PCR Kit (Cat. No. 204243; Qiagen, Hilden, Germany) according to the protocol of the manufacturer. All PCR amplifications were verified via melting curve analysis. The housekeeping gene, namely *β-actin*, was used as an endogenous control to normalize the transcripts’ expression levels. The sequences of the primers utilized in RT-qPCR assays are presented in Table 3. The 2^−ΔΔCT^ method was used to assess the relative mRNA expression outcomes of the examined genes [75].

### 2.8. Histopathological and Immunohistochemical Investigations

At the end of the experimental period, ten birds per group were randomly selected, euthanized by cervical dislocation, and necropsied. Representative tissue samples from the brains and spleens of all birds were harvested, fixed in 10% neutral buffered formalin for 24 h, processed for paraffin technique, sectioned at 5-µm thickness and stained with hematoxylin and eosin [76], and examined microscopically to evaluate any histological alterations. Next, quantitative lesion scoring was carried out in ten randomly selected splenic (10×) microscopic fields per bird. The investigated cerebral lesions were included (neuronal pyknosis, and necrosis, perineuronal vacuolations, gliosis, neuropil vacuolations, vascular congestions, thrombosis, hemorrhages, and leukocytic infiltrations) while the investigated splenic lesions were included, lymphoid depletion, and necrosis, vascular congestions, endothelial hypertrophy, and thrombosis. Finally, the results were expressed as percentages (means ± SEM) using the formula—Lesion FQ (%) = N lesion/N total × 100, where (N lesion) is the number of images exhibited a lesion and (N total) is the total number of images per group.

Successive formalin-fixed, paraffin-embedded, 5 µm thick splenic tissue sections were prepared and immunohistochemically stained following the protocol developed by Hsu, et al. [77]. The splenic tissue sections were stained for (1) TNF-α using the rabbit monoclonal anti-TNF-α primary antibody [TNF/1500R] (ab270264) (abcam, INC, London, UK) at 8 μg/mL dilution with the goat anti-rabbit IgG H&L (HRP) secondary antibody (ab205718) (abcam, INC) at 1/20,000 dilution, and (2) TLR4 using the mouse monoclonal anti-TLR4 primary antibody [76B357.1] (ab22048) (abcam, INC) at (1/100) dilution with the goat anti-mouse IgG H&L (HRP) secondary antibody (ab205719) (abcam, INC) at (1/10,000) dilution. The immune reactions were visualized using 3,3′-Diaminobenzidine (DAB) chromogen and the nuclei were counterstained with Harris hematoxylin. The degree of immunoexpression of the splenic TNF-α and TLR4 were quantified by calculating the percentages of the area fractions of the positively stained brown color concerning the total areas of the images using the image analyzing software, Image J version 1.33, in ten randomly selected (40×) microscopic fields for each marker; the results were expressed as percentages (means ± SEM).

### 2.9. Statistical Analysis of Data

Statistical tests were performed using a one-way analysis of variance (ANOVA) in SPSS version 21 for Windows (SPSS, Inc., Chicago, IL, USA). The regularly distributed nature of the data was established, followed by post hoc Tukey HSD multiple comparisons to determine the statistically significant variations among the various parameters in all experimental replicates. A *p*-value of <0.05 was considered statistically significant. Data were shown as means ± standard error mean (SEM). All graphs were generated using GraphPad Prism 8. Version 8.0.2 (263) (GraphPad Software Inc., San Diego, CA, USA).

## 3. Results

### 3.1. Effect of Dietary Supplementation of CUR, IMI, and Their Combinations on Overall Performance-Related Indices of Ross 308 Broiler Chickens

The results regarding the modulating effects of dietary supplementation of CUR, IMI, and their combinations on the growth performance-related indices (TFI, TBWG, TFCR, and FBW) of Ross 308 broiler chickens are shown in Table 4. The obtained data showed that there were no significant differences (*p* < 0.05) in IBW among all experimental groups. CUR dietary exposure significantly (*p* < 0.05) decreased TFI and TFCR in birds fed CUR by 3.14% and 6.56%, respectively, compared to birds of the C group, but significantly (*p* < 0.05) increased TBWG and FBW by 3.15% and 3.09%, respectively, in birds fed CUR compared to birds of the C group. IMI dietary exposure induced a non-significant (*p* < 0.05) decrease in TFI (0.77% decrease) and a significant increase in TFCR (17.49% increase) in birds fed an IMI-supplemented diet compared to birds of the C group. Additionally, IMI significantly (*p* < 0.05) decreased TBWG and FBW by 15.45% and 15.08%, respectively, in birds fed an IMI-supplemented diet compared to birds of the C group. Supplementation of CUR significantly (*p* < 0.05) decreased TFI by 5.95% and 4.17% in the (CUR+IMI co-treated) and (CUR+IMI pro/co-treated) groups, respectively, and decreased TFCR by 0% and 1.09% in the (CUR+IMI co-treated) and (CUR+IMI pro/co-treated) groups, respectively, compared to birds of the IMI-exposed group.

### 3.2. Effect of Dietary Supplementation of CUR, IMI, and Their Combinations on Blood Hematological-Related Indices of Ross 308 Broiler Chickens

The results regarding the modulating effects of dietary supplementation of CUR, IMI, and their combinations on the blood hematological indices (R.B.Cs, Hb, hematocrit, W.B.Cs, heterophils, and lymphocytes) of Ross 308 broiler chickens are shown in Table 5. The obtained data showed that CUR dietary supplementation induced a significant (*p* < 0.05) decrease in total R.B.C count, Hb content, and Hct value by 11.11, 11.25, and 12.35%, respectively, compared to birds of the C group. IMI dietary exposure induced a significant (*p* < 0.05) decrease in total R.B.Cs count, Hb content, and Hct values by 21.94, 21.22, and 22.58%, respectively, in birds fed IMI when compared to birds of the C group. Supplementation of CUR restored the decrease in total R.B.Cs count, Hb content, and Hct value to decrease of 15.67, 16.78, and 17.74%, respectively, in the (CUR+IMI co-treated) group and to a decrease of 21.37, 19.05, and 21.52%, respectively, in the (CUR+IMI pro/co-treated) group (*p* < 0.05) when compared to birds of the IMI-exposed group.

Dietary supplementation of CUR induced a non-significant (*p* < 0.05) increase of W.B.Cs counts and heterophils % by (25.49 and 0.53% increase, respectively) in birds fed diet supplemented with CUR when compared to birds of the C group. IMI dietary exposure induced a significant (*p* < 0.05) decrease in W.B.Cs counts and lymphocytes % (decrease by 41.57 and 8.57%, respectively) but significantly increased heterophils % (increased by 18.32%) in birds fed IMI-supplemented diet when compared to birds fed control diet. Supplementation of CUR non-significantly (*p* > 0.05) restored the decreased W.B.Cs counts and lymphocytes % (decreased by 16.67 and 4.65%) and restored the heterophils % (increased by 9.96%) in the (CUR+IMI co-treated) birds unlike birds exposed to IMI-and non-treated. Moreover, CUR in the (CUR+IMI pro/co-treated) group, significantly (*p* < 0.05) restored the decreased W.B.Cs count and lymphocytes % to (6.68 and 2.93% decrease) and restored the heterophils % (increased by 6.82%) in comparison with birds exposed to IMI-and non-treated.

### 3.3. Effect of Dietary Supplementation of CUR, IMI, and Their Combinations on Phagocytosis %, Phagocytic Index, Intracellular Killing Capacity, Lysozyme Activity, and IgG of Ross 308 Broiler Chickens

The results regarding the modulating effects of dietary supplementation of CUR, IMI, and their combinations on phagocytosis, phagocytic index, intracellular killing capacity, lysozyme activity, and IgG of Ross 308 broiler chickens are shown in Figure 2. The obtained data showed that CUR dietary supplementation induced a significant (*p* < 0.05) increase in phagocytic % by 17.18% and a non-significant increase in PhI and IKC by 7.62 and 12.67%, respectively, in birds fed a diet supplemented with CUR when compared to birds of the C group. IMI dietary exposure induced a significant (*p* < 0.05) decrease in phagocytic %, PhI, and IKC by 66.67, 49.33, and 56.36% decrease, respectively, in birds fed a diet supplemented with IMI when compared to birds of the C group. Supplementation of CUR significantly (*p* < 0.05) restored the decreased phagocytic %, PhI, and IKC to a decrease of 34.33, 37.22 and 36.63%, respectively, in the (CUR+IMI co-treated) group and to a decrease of 12.12, 13.45, and 21.12%, respectively, in the (CUR+IMI pro/co-treated) group when compared to birds of the IMI-exposed group.

The obtained data showed that dietary supplementation of CUR induced a non-significant (*p* > 0.05) increase of LYZ and IgG serum levels % (increased by 9.07 and 6.05% respectively) in birds fed diet supplemented with CUR if compared to birds of the C group. IMI dietary exposure induced a significant (*p* < 0.05) reduction of LYZ and IgG serum levels (decreased by 47.39 and 38.33% respectively) in birds fed diet supplemented with IMI if compared with birds of the C group. Supplementation of CUR significantly (*p* < 0.05) restored the decreased LYZ and IgG levels to (decreased by 18.86 and 20.25%, respectively) in the CUR+IMI co-treated group and to 16.22 and 19.27% decrease, respectively, in the CUR+IMI pro/co-treated group when compared to birds of the IMI-exposed group.

### 3.4. Effect of Dietary Supplementation of CUR, IMI, and Their Combinations on Serum Biochemical Measurements of Ross 308 Broiler Chickens

The modulatory impact of dietary supplementation of CUR, IMI, and their combinations on blood protein profile and liver function indices (ALT, AST, TP, Album, Gl., and A/G ratio) of Ross 308 broiler chickens is shown in Table 6. The obtained data showed that CUR dietary supplementation induced a significant (*p* < 0.05) decrease in ALT serum levels by 13.74%, but the decrease was non-significant for AST levels (decreased by 19.19%). Also, CUR non-significantly increased TP, albumin, globulin, and A/G ratio by 4.66, 7.79, 0.8 and 9.76%, respectively, in birds fed diet supplemented with CUR when compared to birds of the C group. IMI dietary exposure induced a significant (*p* < 0.05) increase in ALT and AST serum levels (increased by 24.14% and 1-fold, respectively), but significantly decreased TP, albumin, and globulin by 37.28, 39.61, and 34.4%, respectively, in birds fed IMI when compared with birds of C group. Supplementation of CUR significantly reduced the increased AST serum levels by 6.50% and non-significantly (*p* > 0.05) reduced the increased ALT, TP, albumin, globulin, and A/G ratio to 12.05, 27.24, 24.03, 31.20 and 8.13%, respectively, in the (CUR+IMI co-treated) group when compared to birds of the IMI-exposed group. Also, CUR supplementation in the (CUR+IMI pro/co-treated) group significantly (*p* < 0.05) restored the increased ALT and AST levels to 1.59 and 32.66%, respectively, and non-significantly reduced the increased TP, albumin, globulin, and A/G ratio to (decreased to 25.09, 21.43, 29.6 and 15.45%, respectively) unlike birds in the IMI-exposed group.

The modulatory influence of dietary supplementation of CUR, IMI, or their combinations on blood lipid profile (TGs, TC, HDL, LDL, and VLDL) of Ross 308 broiler chickens is shown in Table 6. The obtained data showed that CUR dietary supplementation induced a significant (*p* < 0.05) decrease of TG, TC, LDL, and VLDL serum levels by 44.19, 12.14, 68.01, and 24.53%, respectively, but non-significantly increased HDL serum levels (increased by 5.54%) when compared to the C group. IMI dietary exposure induced a significant (*p* < 0.05) increase of TG, TC, LDL, and VLDL serum levels by 79.17%, 27.46%, 1 fold, and 53.23%, respectively, but significantly decreased HDL serum levels by 29.78% in birds fed IMI when compared with birds of the C group. Supplementation of CUR significantly reduced the increased TG, TC, LDL, and VLDL serum levels to 15.12, 11.27, 32.95, and 5.95%, respectively, and significantly (*p* < 0.05) restored the decreased HDL to 8.50% in the CUR+IMI co-treated group, unlike IMI-exposed group. Also, CUR supplementation in the CUR+IMI pro/co-treated group significantly (*p* < 0.05) reduced the increased TG, TC, LDL, and VLDL serum levels to 3.09, 5.20, 0.20, and 2.58%, respectively and significantly (*p* < 0.05) restored the decreased HDL levels to 2.97% in comparison with birds exposed to IMI.

### 3.5. Effect of Dietary Supplementation of CUR, IMI, and Their Combinations on Oxidative Stress-Related Indices in Spleen of Ross 308 Broiler Chickens

The impact of dietary supplementation of CUR, IMI, or their combinations on serum antioxidant-related parameters (TAC, SOD, CAT, GPx, and MDA) in the spleen of Ross 308 broiler chickens is shown in Figure 3. The obtained data showed that CUR dietary supplementation induced a non-significant (*p* > 0.05) increase of TAC, SOD, CAT, and GPx levels in the spleen by 11.32, 0.59, 7.32, and 18.64%, respectively, and non-significantly decreased MDA levels in the spleen by 9.29% when compared to the C group. IMI dietary exposure induced a significant (*p* < 0.05) decrease of TAC, SOD, CAT, and GPx spleen levels by 54.72, 15.78, 65.06, and 49.73%, respectively, and significantly increased MDA spleen levels by 78.89% in birds fed IMI when compared with birds of the C group. Supplementation of CUR non-significantly elevated the decreased TAC spleen levels by 47.17%. CUR significantly elevated the decreased SOD, CAT, and GPx spleen levels (10.07, 30.12, and 27.12% decrease, respectively) and significantly (*p* < 0.05) reduced the increased MDA levels by 24.89% in the CUR+IMI co-treated group when compared to the IMI-exposed group. Also, CUR supplementation in the CUR+IMI pro/co-treated group significantly (*p* < 0.05) elevated the decreased TAC, SOD, CAT, and GPx spleen levels to 28.30, 5.56, 16.81, and 25.42%, respectively, and significantly (*p* < 0.05) reduced the increased MDA levels to 19.82% when compared to the IMI-exposed group.

### 3.6. Effect of Dietary Supplementation of CUR, IMI, and Their Combinations on mRNA Expression of IL-1β, TLR-4, and IL-10 Genes in Spleen of Ross 308 Broiler Chickens

The effects of dietary supplementation of CUR, IMI, or their combinations on mRNA expression of *IL-1β*, *TLR-4*, and *IL-10* genes in splenic tissues of Ross 308 broiler chickens are shown in Figure 4. The obtained data showed that CUR dietary exposure non-significantly (*p* > 0.05) modulated the *IL-1β* and *TLR-4* mRNA expression by 11 and 6%, respectively, but significantly increased spleen *IL-10* mRNA expression by 17% in the birds fed CUR-supplemented diet compared to the birds of the C group. IMI dietary exposure significantly (*p* < 0.05) up-regulated *IL-1β* and *TLR-4* mRNA expression levels (increased by 1-fold and 1 fold %, respectively), but significantly down-regulated *IL-10* mRNA expression levels (decreased by 74%) in splenic tissues of birds fed diet supplemented with IMI if compared with birds of the C group. Supplementation of CUR significantly (*p* < 0.05) restored the up-regulated *IL-1β* mRNA expression levels to 61%, and non-significantly (*p* > 0.05) restored the up-regulated *TLR-4* mRNA expression levels (increased by 78%). CUR non-significantly restored the down-regulated *IL-10* mRNA expression levels to 61% in splenic tissues in the CUR co-treated group when compared with birds of the IMI-exposed group. Supplementation of CUR significantly (*p* < 0.05) restored the up-regulated *IL-1β* and *TLR-4* mRNA expression levels to 32 and 33%, respectively, and significantly restored the down-regulated *IL-10* mRNA expression levels by 23% in the CUR pro/co-treated group when compared with birds of the IMI-exposed group.

### 3.7. Histopathological and Immunohistochemical Findings

The investigated splenic lesions included lymphoid depletion and necrosis, vascular congestion, endothelial hypertrophy, and thrombosis. The results were expressed as percentages (means ± SEM) and are shown in Table 7. The negative control and CUR-treated groups showed normal histology in Figure 5A, B, whereas the IMI-exposed group manifested some structural alterations such as lymphoid depletion, vascular congestion, and thrombosis (Figure 5C). The response of the splenic tissue to the CUR supplementation in the birds exposed to IMI was reduced in the extent and frequencies of the IMI-induced splenic lesions that were noticed in all CUR-supplemented groups (co-treated or pro/co-treated). The most encountered lesions in these groups were expressed by vascular congestion and lymphoid depletion (Figure 5D, E). The quantitative lesion scoring in the splenic tissue sections of all groups is shown in Table 7.

The data analysis obtained from Image J software (version. 1.32j, http://rsb.info.nih.gov/ij, accessed on 20 November 2023) declared that exposure to IMI significantly upregulated the immunoexpression of the TNF-α and TLR4 in the splenic tissues of the IMID-treated group compared to the control and CUR groups. CUR supplementation either in the CUR co-treated group or CUR pro/co-treated significantly down-regulated the immunoexpression of both the TNF-α and TLR4 in the splenic tissue. The levels of immuno-reactivity of the splenic TNF-α and TLR4 among all groups are shown in Figure 6 and Figure 7, respectively, and statistically summarized in Table 7.

## 4. Discussion

Optimizing the farming practices of chickens is essential for preserving their productivity and assuring the production of high-quality food for human consumption because chickens are a valued source of protein [78,79]. Considering the indiscriminate use of insecticides, birds are frequently exposed to IMI through different sources, and there is a need to search for herbal growth promoters and antioxidative medication to be used in the poultry industry. The purpose of the current study was to assess the impact of dietary supplementation of CUR in ameliorating IMI-induced effects on immune-related indices of Ross 308 broiler chickens.

Regarding the growth performance-related indices, the obtained data revealed that IMI induced a significant increase in TFCR, and significantly decreased TFI, TBWG, and FBW in birds fed the IMI-supplemented diets compared to the C group and the CUR-supplemented group. The obtained outcomes are consistent with those reported by Osman, Shaaban and Ahmed [52], Adegoke et al. [80], and Abotaleb, et al. [81]. In the IMI-fed groups, there was a correlation between the observed lower TFI and lower TBWG and FBW. IMI’s hazardous potential could be the reason for the broilers’ lower feed intake because the pesticides have detrimental toxic effects on the broilers’ gross performance measures, which causes appetite loss and decreased body weight [82]. Additionally, FCR was significantly elevated owing to the reduced weight gain in IMI-exposed birds. Additionally, IMI’s detrimental effects on health status can be related to its effects on the digestion and absorption of feed in the birds’ stomachs, which resulted in poor growth performances [48]. Moreover, IMI exposure resulted in a weakened state of health due to abnormal metabolic and physiological reactions resulting from OS [50,83]. Furthermore, IMI exposure has been linked to a drop in cumulative body gain in birds, as reported by Gibbon et al. [31] and Lopez-Antia et al. [84].

In contrast, birds fed the CUR-supplemented diets in the CUR+IMI co-treated and CUR+IMI pro/co-treated groups showed a significantly decreased TFI and TFCR with an increased TBWG and FBW compared to birds of the C group or the IMI-exposed group. These results were consistent with those reported by Hafez et al. [85], Khodadadi et al. [86], Rajput et al. [87], and Yadav et al. [88]. Hafez et al. [85] reported that CUR dietary supplementation (200 mg or 100 mg) for 42 days significantly enhanced BW and FCR of Cobb 500 chicks during the finisher phase. This may be linked to CUR’s ability to boost bile production, intestinal villi length, cecal width, and gastric digestion fluid, which contributed to better fat digestion. In addition, these advantages were attributed to feeding CUR for a longer period, which improved nutrient absorption during the mature stage [23].

Additionally, CUR boosted intestinal sucrose and maltase activity [89]; up-regulated pancreatic lipase; increased trypsin, amylase, and chymotrypsin; and up-regulated trypsin, amylase, and chymotrypsin. These effects lead to an improved FCR in CUR-supplemented birds. Moreover, the carcass quality was significantly enhanced after adding turmeric powder to the diet of broiler chickens at levels of 500 and 750 mg/kg. In addition, a significant effect of dietary turmeric on BW was discovered at week 3 and later ages concerning desirable biological activities according to Khodadadi et al. [86].

Concerning hematological indices, the obtained results of the current study showed that birds exposed to IMI- had decreased values of RBCs, Hb, Hct, and TLC. These birds showed reduced lymphocytes and increased heterophils. The obtained results are consistent with those of Ravikanth et al. [55], Eid, et al. [90], Sankhala, et al. [91]. Gul et al. [54] reported that birds exposed to IMI showed decreased values of RBCs, Hb, Hct, and WBCs. Oxidative stress, weakened immunity, and metabolic failure in birds exposed to IMI could be responsible for these effects [50,83]. Stressed birds that have been physiologically exposed to insecticides showed disruption in feed digestion and metabolism, which resulted in abnormal values of hematological indices [92]. Additionally, the decline in the mean values of RBC count, Hb, and Hct values may be caused by the direct toxic effects of IMI on the bone marrow, liver, and kidneys, which may have a crucial effect on the production of hemopoiesis and erythropoietin in these organs [55]. The considerable decrease in total leukocytic count observed in the present study was consistent with the results reported by Babu et al. [56] regarding laying birds. The pathological lesions found in the splenic tissues of birds exposed to IMI supported the hypothesis that the lymphocytic depletion seen in the current investigation was caused by splenic hemorrhages.

Regarding the mitigating impact of CUR dietary supplementation, the obtained results showed that birds fed CUR-supplemented diets in the CUR+IMI co-treated or CUR+IMI pro/co-treated groups significantly improved all hematological indices unlike IMI-exposed birds, which are in agreement with those of Adegoke et al. [80], Hafez et al. [85], and Abd El-Samie et al. [93]. This beneficial effect may be linked to CUR’s anti-inflammatory and antioxidant properties, which boosted the metabolism and, in turn, improved iron absorption and utilization, and subsequently RBC formation and Hb concentration. Dietary CUR at the levels of 200 mg or 100 mg for 42 days in Cobb 500 chicks enhanced all hematological indices, along with an improvement in the antioxidant defense system of broiler raised at high stocking density [89]. On the other hand, when broiler birds were supplemented with turmeric powder, Kafi, et al. [94] and Shohe, et al. [95] observed that there were no significant variations in the Hb and PCV levels. Furthermore, broilers fed a basal diet containing 400 g/100 kg CUR exhibited no appreciable changes in PCV, RBCs, or Hb with the exception of WBCs [80]. Ali, et al. [96], demonstrated that a variety of factors, including age, country, temperature, environment, production level, and maintenance system, might affect the variations of erythrocyte count. One type of white blood cell, lymphocytes, can strengthen the immune system and attack pathogens that enter the body [97]. Chickens exposed to excessive stress resulted in higher cortisol hormone production, which in turn resulted in suppression of the immune system, causing lymphoid organs to contract and reduced lymphocyte numbers. In the same line, higher blood levels of cortisol hormone account for the prevention of lymphocyte formation [98,99]. Moreover, Puvadolpirod and Thaxton [99] explained that heat or environmental stress can have an impact on the number of lymphocytes because it reduces the weight of the thymus and bursa Fabricius, two lymphoid organs, which in turn affects the lymphocytes number [100]. In contrast, turmeric extract can serve as an immunomodulator via stimulating lymphocyte formation causing more lymphocyte production [101]. Therefore, in the current investigation, the broilers’ immune system appeared to be strengthened by the increased proportion of lymphocytes, suggesting that dietary CUR can activate T and B lymphocyte cells [102].

Concerning phagocytosis-related indices, the obtained results of the present experiment revealed that birds fed the IMI-supplemented diets had significantly declined Ph % and PhI and IKC values. In contrast, birds fed the diets supplemented with CUR in either the CUR+IMI co-treated group or the CUR+IMI pro/co-treated group showed improved phagocytosis-related indices when compared to IMI-exposed birds. The first immune system activity, known as phagocytic activity, is responsible for engulfing and destroying foreign pathogens before presenting them to the remainder of the immune system for pathogen neutralization [103]. In the present study, IMI dietary exposure markedly decreased Ph % and PhI, indicating IMI-mediated nonspecific adverse effects on cellular immunity. Our outcomes are consistent with those reported by [Abou-Zeid, et al. [104], Mohany, et al. [105], Walderdorff, et al. [106]]. The decrease in phagocytic activity could be brought on by several factors, including the impact of the pesticide used on hematopoietic organs, the uptake prevention of macrophage-arming factor, and an increase in migration inhibitory factor (MIF) brought on by IMI, which inhibits the mobility of neutrophils and macrophages and impairs their capacity to reach inflammatory sites [105]. Additionally, OS may alter receptor-mediated phagocytosis and membrane fluidity; these are essential for innate and adaptive immune responses [107]. Similar results have been reported in domestic chickens exposed to thiamethoxam, a different neonicotinoid, at sub-lethal dosages, which resulted in impaired humoral and cellular immunity [108]. Similar results of decreased phagocytic activity of leukocytes have also has been reported by Mohany et al. [105] and Gawade et al. [109]. In a previous study, old domestic chickens (1–4 weeks old) exposed to IMI at the level of 0.05 mg/kg/d for 37 days experienced a reduction in humoral and cellular immune responses and histopathological changes in the spleen and bursa of Fabricius [39]. Moreover, the results are in agreement with Franzen-Klein, et al. [48]. Neonicotinoid-related immunotoxicity in birds [39,84,110] and the decreased percentages of killed *E. coli* and *C. albicans* after oral exposure may be a sign of immune suppression induced by IMI [48]. Similarly, Khayal, et al. [111] mentioned that IMI decreased phagocytic activity, PhI, and LYZ activity in rats. The observed CUR immune-enhancing effect is consistent with the results described by Farag et al. [112], which may be alluded to the activation of neutrophils and macrophages to produce ROS [113]. Dietary CUR elevated WBCs and phagocytic activity and resulted in greater expression of immune-related cytokines [114].

Humoral and cell-mediated immunity are the two main components of the immune system. According to the current study, there was a significant drop in blood IgG levels of broilers, which may indicate immunosuppression after IMI exposure. Similarly, chickens given IMI showed a marked reduction in serum levels of total immunoglobulin and circulating immune complexes on day 45 [39]. Additionally, the decrease in LYZ activity following IMI exposure is consistent with [Attia, et al. [115], Rahman, et al. [116]]. According to Hafez et al. [85], CUR prevents the decline of Ig synthesis induced by the high stocking density (HSD). Likewise, Isroli, et al. [117] revealed that turmeric could elevate the globulin concentration in broilers. Current outcomes are parallel to the results of Shawky, et al. [118], who stated that turmeric powder (5 g/kg diet) enhanced levels of IgM and IgG in broilers. Similar findings were described by Arshami, et al. [119], who mentioned that the IgM and IgG titers were elevated by various concentrations of CUR powder (0.25 to 0.75%).

Regarding serum biochemical parameters, the obtained results of the present study showed that IMI dietary exposure considerably elevated serum AST and ALT levels. The obtained results are in agreement with those reported by Abu Zeid et al. [41], Eid et al. [90], Mia, et al. [83], Sankhala et al. [91], and Xu et al. [50]. This demonstrates the pathological abnormalities induced by IMI in various essential organs, including the liver and kidneys. Interestingly, Akter, et al. [120] reported that the hepatic serum enzymes AST and ALT were noticeably elevated in rabbits exposed to IMI, which indicated liver injury. Our findings are in the same line with the results reported by Emam et al. [46], who discovered that birds exposed to IMI had higher ALT and AST levels. Lipid peroxidation may be the cause of the unregulated release of ALT and AST after IMI exposure. The overproduction of ROS implicated in OS caused by pesticide exposure leads to increased MDA levels [52]. When MDA levels are higher than the capacity of the body’s defenses against oxidation, ROS severely damage cellular DNA [121]. Therefore, increased MDA levels may be responsible for the change in the metabolism, physiological responses, immunological responses, and growth rate of birds exposed to IMI poisoning. The outcomes of this study revealed that birds exposed to IMI had elevated MDA levels. Indeed, IMI-induced ROS overproduction leads to high LPO and inflammatory responses [122]. CUR supplementation in the diet reduced ALT and AST activities in birds, demonstrating CUR’s capacity to scavenge and neutralize free radicals to protect liver cells from attack by free radicals [85,123]. The observed decline in liver enzymes’ activities in the present experiment confirmed the improved liver functions and showed the antioxidant-modulating effect of CUR supplementation and its role in preventing liver cell damage induced by free radicals.

The drop in albumin, globulin, and total protein levels found in broilers’ blood after IMI exposure was parallel to results reported by Abu Zeid et al. [41]. This might be caused by competitive inhibition of phenylalanine-t-RNA synthesis, which prevents amino-acylation and peptide elongation and inhibits hepatic protein synthesis at the post-transcriptional stage [124]. Reducing the downstream absorption of digested amino acids from birds’ guts to the blood is considered one of the IMI-induced impacts on blood protein reductions [47,48]. Additionally, Nebbia [125] reported that the liver tissue predominantly detoxifies and eliminates xenobiotics that are harmful to birds.

The observed effect of IMI in increasing TC in our investigation is consistent with findings showed that stressors elevate blood cholesterol concentrations in broiler chickens. In comparison to birds fed a control diet, we noticed that adding CUR to broiler chickens’ diets reduced TG, TC, LDL-C, and VLDL-C levels, while increasing HDL levels. Hafez et al. [85] discovered that feeding broilers CUR lowered their blood’s TC compared to those fed a control diet. Furthermore, it has been reported the impact of dietary CUR supplementation on TC and LDL in rats [126].

Notably, IMI induced a decrease in SOD, CAT, GPx, and TAC levels; this is consistent with Zhang, et al. [127], who found that the concentrations of T-AOC, SOD, GSH, and GSH-Px were reduced, but the levels of MDA were elevated in renal tissue of mice given (10 mg/L,) IMI for 30 days in drinking water when compared with the control mice. This is supported by the fact that CUR therapy reversed the elevated MDA levels and decreased T-AOC, SOD, GSH, and GSH-Px concentrations in the current study in contrast, exposure to IMI reduced the antioxidants levels, resulting in an accumulation of lipid peroxide, which led to inflammatory reactions.

According to Kammon et al. [39], exposure of birds to IMI had negative effects on biochemical and immune system indices and induced OS-related biomarkers. The hepatic tissue of laying birds fed IMI showed a considerable decline in GSH levels and raised levels of thiobarbituric acid reactive compounds (TBARS) [56]. In RBCs, liver, kidney, and testes of cockerels, IMI induced oxidative stress, reduced CAT, and increased lipid peroxidation [59]. In agreement with Ravikanth et al. [61] and Subha [128], IMI-exposed birds had significantly lower levels of antioxidant-related biomarkers and higher lipid peroxidation levels in the splenic tissue.

Similar to our findings, Salah et al. [89] reported that dietary supplementation of CUR in chickens reduced the TBA (thiobarbituric acid) and enhanced the activities of antioxidant enzymes. CUR prevents OS by increasing the antioxidant enzyme activity including SOD, CAT, and GPx enzymes. On the other hand, CUR could antagonize OS induced by IMI in broilers’ spleen, which is consistent with the findings of Cheng, et al. [60]. Similarly, Li, et al. [129], reported that CUR mitigated the AFB1-induced changes in GSH, SOD, CAT, and MDA levels. This might be a result of CUR’s capacity to neutralize free radicals by boosting the activities of antioxidant enzymes and decreasing OS [130]. Additionally, earlier studies reported by [Nayak and Sashidhar [131], Zhang, et al. [132]] have shown that CUR has antioxidant properties against hepatic damage induced by AFB1in rats and chicks.

In the current investigation, IMI-exposed broilers showed up-regulated expression levels of *IL1β* and *TLR-4* in splenic tissues and, in contrast, down-regulated *IL-10* mRNA expression levels. The spleen is a peripheral lymphoid organ that plays a crucial role in the body’s innate and adaptive immune responses against systemically acquired antigens [133,134].To the best of our knowledge, this is the first study that reported that IMI-induced up-regulated the expression level of the *TLR-4* gene in the spleen of IMI-exposed broilers. Similar findings were obtained by Farag, et al. [135], who found that the Toll-like receptor 2 (*TLR2*) gene was up-regulated in the brains of rats after IMI exposure, while *TLR2* had a strong immunopositive response in the brains of IMI-treated rats. Toll-like receptor is activated by molecules generated from pathogens, which causes the production of inflammatory mediators including TNF-α [136]. Lipopolysaccharide (LPS), a crucial part of Gram-negative bacteria’s cell walls, ligates *TLR4* to start an inflammatory response in organs [137]. There is evidence that IL-1β, TNFα, and interferon (INFγ) expression levels are increased in the blood and brain after sub-lethal exposure to insecticides such as acephate [138]. The identification of endogenous damage-associated molecular patterns (DAMPs) and exogenous pathogen-associated molecular patterns (PAMPs) is a critical function of *TLRs* [139]. Stimulation of *TLRs* by PAMPs or DAMPs results in the release of chemokines and cytokines, which intensifies inflammatory responses [140]. Inflammation and OS have complicated interactions. Cell injury as a result of redox stress brought on by ROS can activate *TLR2/4* and the downstream gene *NF-B*, producing inflammation and organ damage [141]. Our obtained outcomes are in disagreement with those of Pandit et al. [142], who reported that there was a non-significant fold increase in the expression of *TLR-4* and *TNFα* mRNA following the oral administration of IMI.

Similarly, Farag, et al. [143] observed that oxidative damage was significantly enhanced by thiacloprid, including elevated protein carbonyl, MDA content, and DNA damage in the brain of exposed chicken embryos. Additionally, proinflammatory cytokine induction is a primary indicator of the inflammatory process [143]. High levels of nitric oxide (NO) can trigger pro-inflammatory cytokine (*IFN-γ*, *TNF-α*, and *IL-1β*) overexpression. According to the current research, IMI exposure caused splenic tissues to become oxidatively damaged, which, in turn, induced or encouraged subsequent inflammatory responses that were probably mediated by the NF-κB signaling system. Herein, the observed up-regulation of *IL-1β* mRNA expression served as evidence for this hypothesis.

Curcumin supplementation markedly decreased the mRNA expression patterns of *IL-1β*, mucin 2, *COX-2*, and *p38 MAPK* in the ileal mucosa [144] and serum *TNF-α* concentration [145]. CUR predominantly alters the *p38-MAPK* pathway and thus suppresses the downstream activation of *IL-1β*, *IL-6*, and *TNF-α* genes.

The histopathological alterations observed in the current study were consistent with those of Gupta and Deepika [146], who reported that 14 day old broiler chicks exposed to IMI showed congestion in the spleen. Additionally, splenic tissues of rats treated with IMI displayed low numbers of lymphocytes, and thymus tissues showed lymphocytic depletion with pyknotic nuclei [105,147]. The intense immunopositive reactivity of TLR2 and TNFα in the liver found in this study could be explained by the up-regulation of TLR4 and TNFα in the spleen in this study. The obtained results are in agreement with those of Alhusaini, et al. [148], who reported that acetamiprid-treated rats showed detectable TLR4 protein expression in the glomeruli that were mainly localized to podocytes, moderately expressed along the proximal tubule and strongly expressed along the distal tubule

## 5. Conclusions

Herein, the current study provided potent evidence that the consumption of CUR improved the broiler chicken’s immune system. Curcumin counteracts IMI-induced detrimental effects on growth performance, hematological associated indices, oxidative status, and mRNA expression of immune-related genes (*IL-1β*, *TLR-4*, and *IL-10*) in 308 Ross broiler chickens. Thus, CUR has the potential for utilization as a remedy in biomedical and pharmaceutical products, especially as antioxidant supplements constructing a precise antidote owing to its unique properties for numerous biomedical applications. Finally, our results provide new insight into IMI-induced immuno-toxicity in broiler chickens, moreover, for the first time, CUR preserves an in vivo protective effect against IMI principally in the pro/concurrent-supplementation compared with the con-current supplementation only.

## Figures and Tables

**Figure 1 animals-13-03650-f001:**
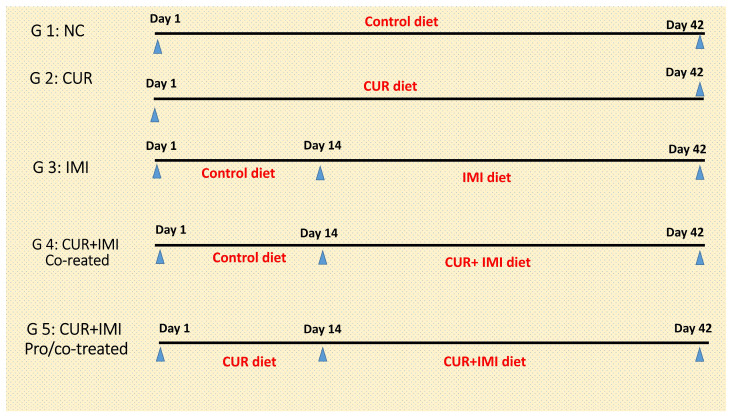
Experimental groups, doses, durations, and treatments: CUR (450 mg/kg diet) and IMI (50 mg/kg diet) in Ross 308 broilers.

**Figure 2 animals-13-03650-f002:**
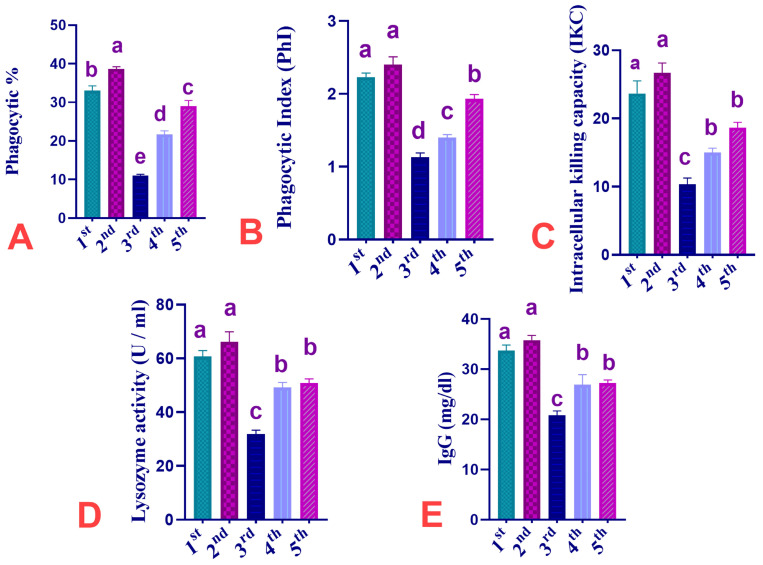
Effect of dietary supplementation of CUR, IMI, and their combinations on Phagocytosis, lysozyme activity, and IgG of Ross 308 broiler chickens. 1st group C, 2nd group CUR, 3rd group IMI-treated, 4th group CUR+IMI Co-treated, 5th group CUR+IMI Pro/co-treated. (**A**): Phagocytosis % (Ph %). (**B**): Phagocytic index (PhI). (**C**): Intracellular killing capacity (ICK). (**D**): Lysozyme activity (LYZ). (**E**): Immunoglobulin G (IgG). Values are shown as mean ± SEM of (6) birds per experimental group. Means bearing different superscripts are significantly different at *p* < 0.05.

**Figure 3 animals-13-03650-f003:**
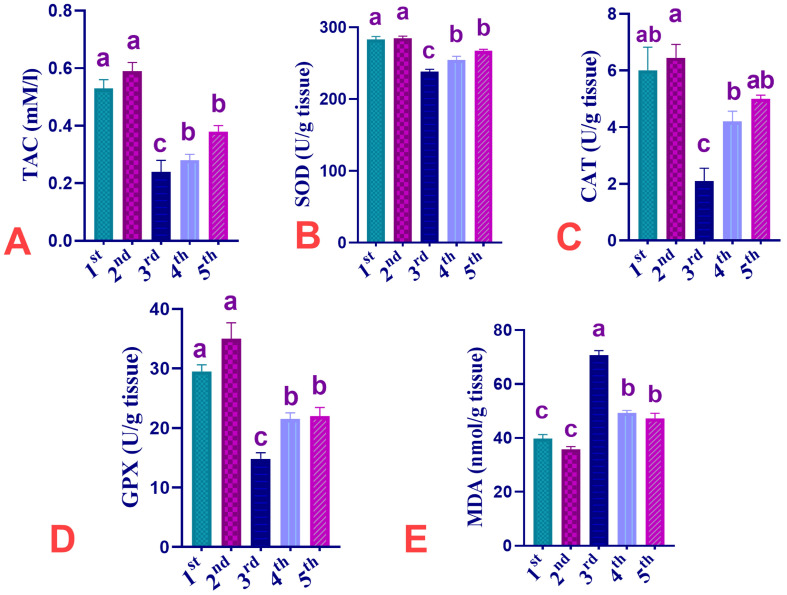
Effect of dietary supplementation of CUR, IMI, and their combinations on Phagocytosis, lysozyme activity, and IgG of Ross 308 broiler chickens. 1st group C, 2nd group CUR, 3rd group IMI-treated, 4th group CUR+IMI Co-treated, 5th group CUR+IMI Pro/co-treated. (**A**): Total antioxidant capacity (TAC). (**B**): Superoxide dismutase (SOD). (**C**): Catalase (CAT). (**D**): Glutathione peroxidase (GPx). (**E**): Malondialdehyde (MDA). Values are shown as mean ± SEM of (6) birds per experimental group. Means bearing different superscripts are significantly different at *p* < 0.05.

**Figure 4 animals-13-03650-f004:**
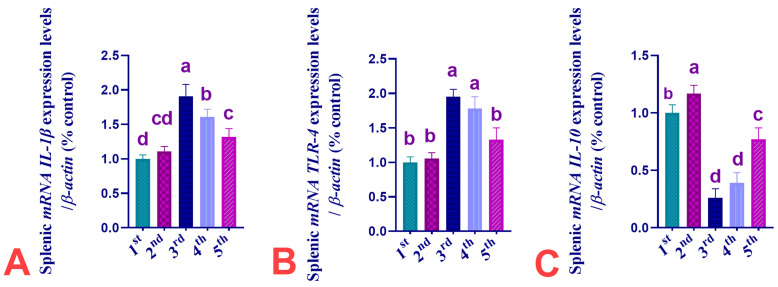
Effect of dietary supplementation of CUR, IMI, and their combinations on *mRNA* expression of *IL-1β*, *TLR-4*, and *IL-10* genes in spleen of Ross 308 broiler chickens. 1st group C, 2nd group CUR, 3rd group IMI-treated, 4th group CUR+IMI Co-treated, 5th group CUR+IMI Pro/co-treated. (**A**): Interleukin 1 Beta (*IL-1β*). (**B**): Toll-Like Receptor 4 (*TLR-4*). (**C**): Interleukin 10 (*IL-10*). Values are shown as mean ± SEM of three birds per experimental group. Means bearing different superscripts are significantly different at *p* < 0.05.

**Figure 5 animals-13-03650-f005:**
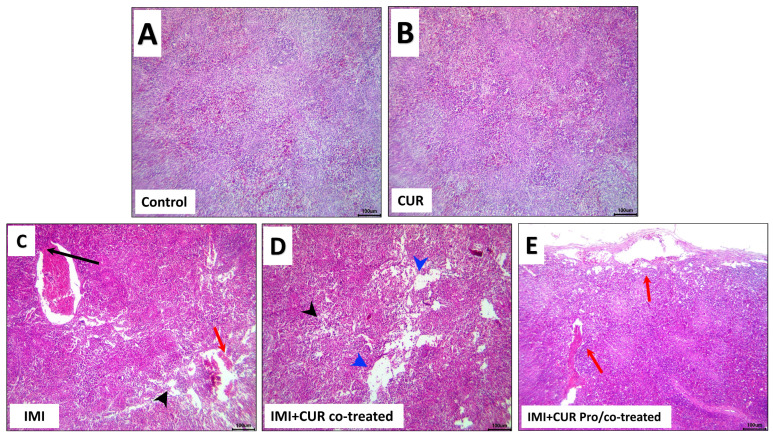
Representative photomicrographs of H&E-stained splenic tissue sections for the effect of dietary supplementation of CUR, IMI, and their combinations splenic tissue structure showing normal histology in the control (**A**) and the CUR (**B**) groups. Histopathological alterations in the IMI-treated group (**C**), with reduction of these splenopathic alterations in the CUR co-treated group or CUR pro/co-treated group. (**D**,**E**). The symbols in the images denote black arrowhead; lymphoid depletion, blue arrowhead; lymphoid necrosis, black arrow; thrombus, red arrow; vascular congestion. (Scale bar is 100 μm).

**Figure 6 animals-13-03650-f006:**
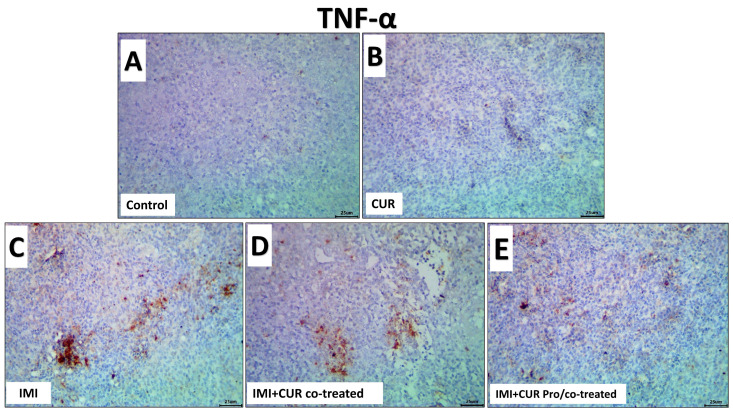
Representative photomicrographs of the TNF-α immuno-stained splenic tissue sections for the effect of dietary supplementation of CUR, IMI, and their combinations in spleen of Ross 308 broiler chickens. (**A**,**B**): control and CUR groups for TNF-α showing weak expression in splenic tissue. (**C**): showing up-regulation of the TNF-α expression in the IMI-treated group in splenic tissue. (**D**,**E**): CUR-supplemented groups, showed down-regulation of TNF-α expression in the CUR co-treated group or CUR pro/and co-treated groups. (Scale bar is 25 μm).

**Figure 7 animals-13-03650-f007:**
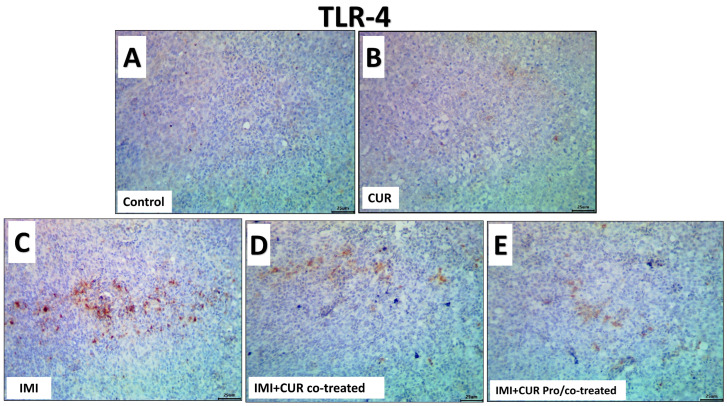
Representative photomicrographs of the TLR-4 immuno-stained splenic tissue sections for the effect of dietary supplementation of CUR, IMI, and their combinations in spleen of Ross 308 broiler chickens. (**A**,**B**): control and CUR groups for TLR-4 showing weak expression in splenic tissue. (**C**): showing up-regulation of the TLR-4 expression in the IMI-treated group in splenic tissue. (**D**,**E**): CUR- supplemented groups, showed down-regulation of TLR-4 expression in the CUR-treated group or CUR pro/and co-treated groups. (Scale bar is 25 μm).

**Table 1 animals-13-03650-t001:** The ingredients and nutrient contents of basal diet.

Ingredient, %	Starter (1–10 Days)	Grower (11–20 Days)	Finisher (21–35 Days)
Yellow corn	57.8	61	65.2
Soybean meal, 48%	34.8	30.3	25.6
	1.2	1.2	1.2
Soybean oil	0	0	0
Calcium carbonate	1.8	3.2	3.7
Calcium diphasic phosphate	1.2	1.2	1.2
Common salt	1.2	1.2	1.2
Premix ^1^	0.3	0.3	0.3
L-Lysine HCL, 78%	0.8	0.8	0.8
DL-Methionine, 99%	0.35	0.3	0.3
Choline chloride	0.20	0.20	0.20
Anti-mycotoxin	0.10	0.10	0.10
Calculated composition			
Metabolizable energy (Kcal/Kg)	3013	3130	3200
Crude protein, %	23.02	21.00	19.22
Ether extract, %	4.31	5.76	6.35
Crude fiber, %	2.64	2.54	2.47
Calcium, %	1.09	1.08	1.07
Available phosphorous, %	0.47	0.44	0.48
Lysine, %	1.45	1.29	1.17
Methionine, %	0.60	0.53	0.51

^1^ Vitamin premix supplied per kilogram of diet: tocopheryl acetate, 80 mg; cholecalciferol, 5000 IU; retinol, 20.000 IU; menadione, 2.9 mg; thiamine, 4.5 mg; riboflavin, 5.5 mg; pantothenate, 10 mg; niacin, 450 mg; folate, 2.5 mg; pyridoxine, 8 mg; biotin, 250 μg; Fe (sulphate), 40 mg; cyanocobalamine, 17 μg; Se (selenate), 0.3 mg; I (iodide), 1.50 mg; Cu (sulphate), 8 mg; Zn (sulphate and oxide), 120 mg; Mn (sulphate and oxide), 120 mg.

**Table 2 animals-13-03650-t002:** Chemical composition (%) of feedstuffs used in formulation of the experimental diets (air dry basis).

Ingredient	Nutrient (%)								
	Moisture	CP	EE	* CF	* Calcium	* AP ^1^	* Lysine	* Methionine	* ME ^2^
yellow corn	8	8.9	3.7	2.3	0.05	0.08	0.26	0.18	3350
Soybean meal, 48%	10.6	47.8	1.1	3.7	0.25	0.27	2.92	0.67	2440
Corn gluten, 60%	11	59.4	2.4	1.8	0.07	0.14	1.03	1.78	3720
Soybean oil	-	-	98	-	-	-	-	-	8800
Calcium carbonate	-	-	-	-	38	-	-	-	-
Calcium dibasic phosphate	-	-	-	-	26	18	-	-	-
premix	-	-	-	-	26	-	-	-	-
Lysine, HcL, 78%	-	118	-	-	-	-	78	-	4600
DL-Methionine, 98%	-	58	-	-	-	-	-	98	3600

^1^ Available phosphorus; ^2^ Metabolizable energy Kcal/kg; * Calculated according to Council [66]. Moisture, CP (crude protein), EE (ether extract), CF (crude fiber) were chemically analyzed according to procedures of AOAC [67].

**Table 3 animals-13-03650-t003:** Primer Oligonucleotide Sequences for target gene RT-PCR analysis.

Gene	Forward Primer (5′–3′)	Reverse Primer (5′–3′)	Gene Bank Accession No.	Product Size
TLR-4	GTTCTTCTGTGACCCGTGAGA	GTGAGGAGCGTTGCGCTTT	FJ915520.1	129
IL-1β	TGCCTGCAGAAGAAGCCTCG	CTCAGGTCGCTGTCAGCAAAG	NM_204524.2	173
IL-10	TTGGGGTGGCATTTCTCCTTG	GTTAGACTGCCTCAAACAGCG	EU999771.1	89
actin-b	GTGGATCAGCAAGCAGGAGT	ATCCTGAGTCAAGCGCCAAA	NM_205518.2	182

*TLR-4*: Toll-Like Receptor 4, *IL-1β*: Interleukin 1 Beta, *IL-10*: Interleukin 10.

**Table 4 animals-13-03650-t004:** Effect of dietary supplementation of imidacloprid (IMI), curcumin (CUR), and their combinations on overall performance of Ross 308 broiler chickens.

Parameter	IBW (g)	TFI (g)	TBWG (g)	TFCR	FBW (g)
Group					
1st group C	45.20 ± 0.80	4300 ± 8.097 ^a^	2349 ± 8.84 ^b^	1.83 ± 0.01 ^b^	2394 ± 8.79 ^b^
2nd group CUR	46.00 ± 0.84	4133 ± 18.25 ^b^	2423 ± 9.67 ^a^	1.71 ± 0.01 ^c^	2468 ± 9.50 ^a^
3rd group IMI-treated	44.00 ± 1.30	4267 ± 11.40 ^a^	1986 ± 36.46 ^d^	2.15 ± 0.04 ^a^	2033 ± 36.17 ^e^
4th group CUR+IMI Co-treated	44.20 ± 0.86	4013 ± 23.74 ^c^	2196 ± 2.88 ^c^	1.83 ± 0.01 ^b^	2242 ± 2.59 ^d^
5th group CUR+IMI pro/co-treated	45.40 ± 0.51	4089 ± 12.20 ^b^	2259 ± 5.51 ^c^	1.81 ± 0.01 ^b^	2303 ± 5.24 ^c^

Total feed intake (TFI), Total body weight gain (TBWG), Total feed conversion ratio (TFCR), and Final body weight (FBW). Values are mean ± SEM of six birds per experimental group. Means within the same column carrying different superscripts were significantly different at (*p* < 0.05).

**Table 5 animals-13-03650-t005:** Effect of dietary supplementation of imidacloprid (IMI), curcumin (CUR), and their combinations on blood hematological-related indices of Ross 308 broiler chickens.

Parameters	RBCs (×10^6^ mL^3^)	Hb (g/dL)	Hct (%)	TLC (×10^9^/L)	Heterophils %	Lymphocytes %
Group						
1st group C	3.51 ± 0.13 ^a^	10.13 ± 0.34 ^a^	31.00 ± 0.73 ^a^	10.20 ± 1.62 ^abc^	31.83 ± 1.14 ^ab^	68.17 ± 1.14 ^a^
2nd group CUR	3.12 ± 0.02 ^b^	8.99 ± 0.04 ^b^	27.17 ± 0.11 ^b^	12.80 ± 0.42 ^a^	32.00 ± 0.58 ^a^	68.00 ± 0.58 ^a^
3rd group IMI-treated	2.74 ± 0.14 ^d^	7.98 ± 0.29 ^b^	24.00 ± 0.84 ^c^	5.96 ± 0.63 ^d^	37.66 ± 1.02 ^c^	62.33 ± 1.02 ^b^
4th group CUR+IMI Co-treated	2.96 ± 0.02 ^bc^	8.43 ± 0.05 ^b^	25.50 ± 0.18 ^ab^	8.50 ± 0.55 ^bcd^	35.00 ± 0.68 ^bc^	65.00 ± 0.68 ^ab^
5th group CUR+IMI Pro/co-treated	2.76 ± 0.10 ^bc^	8.20 ± 0.12 ^b^	24.33 ± 0.59 ^c^	9.50 ± 0.37 ^bc^	33.83 ± 1.11 ^b^	66.17 ± 1.11 ^a^

Red blood cells (R.B.Cs.); hemoglobin (Hb), hematocrit value (Hct), total leukocytic count (TLC). Values are mean ± SEM of six birds per experimental group. Means within the same column carrying different superscripts were significantly different at (*p* < 0.05).

**Table 6 animals-13-03650-t006:** Effect of dietary supplementation of imidacloprid (IMI), curcumin (CUR), and their combinations on serum biochemical measurements (liver function, protein profile, and lipid profile) of Ross 308 broiler chickens.

Parameter	ALT (U/L)	AST (U/L)	TP (g/dL)	Albumin (g/dL)	Globulin (g/dL)	A/G Ratio	TG (mg/dL)	TC (mg/dL)	HDL (mg/dL)	LDL (mg/dL)	VLDL (mg/dL)
Group											
1st group C	20.67 ±0.92 ^b^	215.33 ±24.58 ^b^	2.79 ±0.16 ^ab^	1.54 ±0.09 ^a^	1.25 ±0.07 ^a^	1.23 ±0.01	43.00 ±1.32 ^b^	115.33 ±3.75 ^c^	78.33 ±1.12 ^ab^	34.60 ±1.27 ^c^	10.07± 0.22 ^b^
2nd group CUR	17.83 ±0.60 ^c^	174.00 ±20.10 ^b^	2.92 ±0.17 ^a^	1.66 ±0.07 ^a^	1.26 ±0.11 ^a^	1.35 ±0.10	24.00 ±0.73 ^c^	101.33 ±2.74 ^d^	82.67 ±1.65 ^a^	11.07 ±1.00 ^d^	7.60± 0.48 ^b^
3rd group IMI-treated	25.66 ±0.56 ^a^	438.33 ±55.56 ^a^	1.75 ±0.09 ^b^	0.93 ±0.04 ^d^	0.82 ±0.05 ^b^	1.16 ±0.05	77.04 ±2.01 ^a^	147.00 ±4.21 ^a^	55.00 ±2.39 ^c^	76.57 ±1.44 ^a^	15.43± 0.39 ^a^
4th group CUR+IMI Co-treated	23.16 ±0.70 ^ab^	229.33 ±8.83 ^b^	2.03 ±0.11 ^b^	1.17 ±0.08 ^cd^	0.86 ±0.05 ^b^	1.33 ±0.08	49.50 ±1.78 ^b^	128.33 ±2.79 ^b^	71.67 ±4.59 ^b^	46.00 ±4.05 ^b^	10.67± 0.22 ^b^
5th group CUR+IMI Pro/co-treated	21.00 ±0.73 ^b^	285.67 ±16.61 ^b^	2.09 ±0.16 ^bc^	1.21 ±0.09 ^bcd^	0.88 ±0.08 ^b^	1.42 ±0.11	44.33 ±2.76 ^b^	121.00 ±1.32 ^bc^	76.00 ±2.03 ^ab^	34.67 ±3.17 ^c^	10.33± 0.28 ^b^

Alanine aminotransferase (ALT), Aspartate aminotransferase (AST), Total protein (TP), Albumin (Album), Globulin (Gl.), Albumin/Globulin ratio (A/G ratio), Triglycerides (TGs), Total Cholesterol (TC), High-density lipoprotein (HDL), Low density lipoprotein (LDL), and Very low-density lipoprotein (VLDL). Values are mean ± SEM of six birds per experimental group. Means within the same column carrying different superscripts were significantly different at (*p* < 0.05).

**Table 7 animals-13-03650-t007:** Effect of dietary supplementation of IMI, CUR, and their combinations on the splenic histology, and the immuno-expression of the splenic TNF-α and TLR-4 of Ross 308 broiler chickens.

Groups	1st Group C	2nd Group CUR	3th Group IMI-treated	4th Group CUR+IMI Co-Treated	5th Group IMI+CUR Pro/Co-Treated
Parameters					
Lymphoid depletion	0.00 ± 0.00 ^c^	0.00 ± 0.00 ^c^	32.00 ± 2.91 ^a^	8.00 ± 2.00 ^b^	9.00 ± 1.79 ^b^
Lymphoid necrosis	0.00 ± 0.00 ^b^	0.00 ± 0.00 ^b^	6.00 ± 1.63 ^a^	2.00 ± 1.33 ^ab^	4.00 ± 1.63 ^ab^
Vascular congestion	0.00 ± 0.00 ^c^	0.00 ± 0.00 ^c^	16.00 ± 1.63 ^a^	5.00 ± 1.66 ^bc^	7.00 ± 1.52 ^b^
Vascular thrombosis	0.00 ± 0.00 ^a^	0.00 ± 0.00 ^a^	2.00 ± 1.33 ^a^	0.00 ± 0.00 ^a^	1.00 ± 0.10 ^a^
Endothelial hypertrophy	0.00 ± 0.00 ^a^	0.00 ± 0.00 ^a^	5.00 ± 1.66 ^a^	2.00 ± 1.33 ^a^	3.00 ± 2.13 ^a^
TNF-α Immuno-positive area fraction	0.48 ± 0.06 ^c^	0.47 ± 0.04 ^c^	7.49 ± 0.43 ^a^	3.63 ± 0.13 ^b^	4.38 ± 0.22 ^b^
TLR4 immuno-positive area fraction	0.43 ± 0.07 ^d^	0.46 ± 0.08 ^d^	8.18 ± 0.44 ^a^	4.01 ± 0.11 ^bc^	4.62 ± 0.14 ^b^

Values are mean ± SEM of 10 birds per experimental group. Means within the same row carrying different superscripts were significantly different at (*p* < 0.05). TNF-α and TLR-4 splenic immuno-reactivity.

## Data Availability

The data depicted in this work are available upon asking from the corresponding author.

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
