# Peer review of "Dietary Curcumin Modulating Effect on Performance, Antioxidant Status, and Immune-Related Response of Broiler Chickens Exposed to Imidacloprid Insecticide"

_animals, 2023, doi:10.3390/ani13233650_

Round 1

Reviewer 1 Report

Comments and Suggestions for Authors

The manuscript entitled “Dietary Curcumin Modulating Effect on: Performance, Antioxidant Status, and Immune- Related Response of Broiler Chickens Exposed to Imidacloprid Insecticide” reported the protective effects of curcumin (CUR) supplementation on growth performance, anti-oxidative capacity, immune to hypertrophy (IMI) exposed birds. In general the experiment was well designed and the content is of some scientifically intriguing. However, I would have a few details and questions to the author before it can be considered for publication.

Commend to the author

1.L36-40, L58-61: The language should be concise or reorganized, and the language of the entire manuscript should be improved.Avoid too long sentence, also the background and objective in the abstract is confusion!

2.The writing in poor and need improve, For example : Line 415-417. The obtained data showed that CUR dietary supplementation induced a non-significant (P < 0.05) increase of LYZ and IgG serum levels % by (9.07 and 6.05 % increase) respectively, in birds fed diet supplemented with CUR if compared to birds of the C group. What it mean non-significant (P < 0.05) ?    change into Dietary supplementation with CUR increased (P>0.05) serum LYZ and IgG levels by 9.07 and 6.05 % , respectively, when compared with that of the C group. the author should present results clear and concise across the text!

3. P < 0.05, P should in Italic,check through the text

4. L175: “…were done in accordance with [55] shown in…”, can references be inserted directly into a sentence? Please use proper citation style in the manuscript.

5.Literature citation should follow the journals style (eg Line 648-649, Line Line 653-654,...). List the first author and the publish year, eg Hafez et al (2022) reported that ....[76]. check across the text!

6.L482: According to the author’s description, the antioxidant stress-related indicators were measured in spleen, not serum. This is inconsistent with the description in the materials and methods (L259-267).

7. spelling and grammar check is required before publication.

Comments on the Quality of English Language

spelling and grammar check is required before publication.

Author Response

Comment 1: L36-40, L58-61: The language should be concise or reorganized, and the language of the entire manuscript should be improved. Avoid too long sentence, also the background and objective in the abstract is confusion!

Response 1: Many thanks for pointing this out towards improving our manuscript. All the manuscript was edited for English language, writing or grammatical errors by MDPI. Assignment number: english-73492. The certificate for MDPI is uploaded. It was done and corrected as recommended.

Comment 2: The writing in poor and need improve, For example:” Line 415-417. The obtained data showed that CUR dietary supplementation induced a non-significant (P < 0.05) increase of LYZ and IgG serum levels % by (9.07 and 6.05 % increase) respectively, in birds fed diet supplemented with CUR if compared to birds of the C group”. What it mean “non-significant (P < 0.05) “? Change into “Dietary supplementation with CUR increased (P>0.05) serum LYZ and IgG levels by 9.07 and 6.05 %, respectively, when compared with that of the C group”. The author should present results clear and concise across the text!

Response 2: Many thanks for your valuable comment. It was done and corrected in the manuscript as recommended.

Comment 3: P < 0.05, P should in Italic, check through the text

Response 3: Many thanks for your valuable comment. It was done and corrected in the manuscript as recommended.

Comment 4: L175: “…were done in accordance with [55] shown in…”, can references be inserted directly into a sentence? Please use proper citation style in the manuscript.

Response 4: Many thanks for your helpful suggestion. All the cited references were revised using the Endnote 7 introducing the proper citation style of the journal and corrected as recommended by reviewer.

Comment 5: Literature citation should follow the journal’s style (eg Line 648-649, Line 653-654,..). List the first author and the publish year, eg Hafez et al (2022) reported that ...[76]. Check across the text!

Response 5: Many thanks for your helpful suggestion. All the cited references were revised using the Endnote 7 introducing the proper citation style of the journal and corrected as recommended by reviewer.

Comment 6: L-482: According to the author’s description, the antioxidant stress-related indicators were measured in spleen, not serum. This is inconsistent with the description in the materials and methods (L259-267).

Response 6: Many thanks for your thoughtful comment. Spleen homogenates were used for estimation of oxidative stress-related indices cited at lines 221-225 for sample preparation and at lines 263-269 for methods of analysis. Additionally serum samples were used for estimation of all other biochemical indices at lines 214-217 for serum collection, and at lines 249-262 for methods of analysis.

Comment 7: Spelling and grammar check is required before publication.

Response 7: Done as recommended. All the manuscript was revised for any writing or grammatical errors and were corrected.

Comments on the Quality of English Language: Spelling and grammar check is required before publication.

Response 8: Thank you for your valuable comment, the manuscript has been checked for grammatical errors and the all corrections were done as recommended and highlighted in the manuscript. The manuscript was checked for language and grammar by MDPI. Assignment number: english-73492. The certificate for MDPI is uploaded.

Please address all correspondence concerning this manuscript to me at

[email protected]

Thank you for your consideration of this manuscript.

Sincerely, Ehsan H. Abu-Zeid

Reviewer 2 Report

Comments and Suggestions for Authors

Dear Authors,

The Manuscript entitle: " Dietary Curcumin Modulating Effect on: Performance, Antioxi- 2 dant Status, and Immune- Related Response of Broiler Chickens 3 Exposed to Imidacloprid Insecticide"  is very innovative and addresses the incorporation of turmeric in chicken feed in a rigorous way. As a curiosity I would like to know about the temperature and humidity ranges in which the test was carried out and if the animals had environmental entertainment measures, please include this information in the manuscript.

Best Regards

Author Response

Comment 1: The Manuscript entitle: "Dietary Curcumin Modulating Effect on: Performance, Antioxidant Status, and Immune-Related Response of Broiler Chickens Exposed to Imidacloprid Insecticide" is very innovative and addresses the incorporation of turmeric in chicken feed in a rigorous way. As a curiosity I would like to know about the temperature and humidity ranges in which the test was carried out and if the animals had environmental entertainment measures, please include this information in the manuscript.

Response 1: Many thanks for pointing this out towards improving our manuscript.

Birds were reared in a naturally ventilated open house in animal research unit of Faculty of Veterinary Medicine, Zagazig University.

Birds were reared in batteries with automatic water system, ban feed in front of birds and at a density 10 birds/m2. Lighting regime was 24 h from days 1 to 3 and then 23 h lighting was applied up to the end of the experiment. The starting temperature was adjusted to 33 ± 1 â—¦C for the first 3 days, and then decreased by 3â—¦C each week until it reached 24â—¦C at the end of the experimental period according to the Aviagen guidelines (Aviagen, 2018). Humidity was maintained around 60% throughout the whole experiment. Information was included in the manuscript at lines 171-175.

Please address all correspondence concerning this manuscript to me at

[email protected]

Thank you for your consideration of this manuscript.

Sincerely, Ehsan H. Abu-

Reviewer 3 Report

Comments and Suggestions for Authors

1.      The research is very field oriented and applicable.

2.      The use of colon (:) in title is unnecessary.

3.      Abstract: The first sentence (line 36 to 41) of abstract is too lengthy and not elaborative and need improvement to make the sense clearer.  

4.      Abstract: the use “neonicotinoids” class of insecticide and then using the precise member of class name “imidacloprid” in same sentence can be confusing, so I will suggest using “imidacloprid” only.

5.      The article is lots of language issue, which need serious consideration.

6.      The authors fail to establish the sense of inclusion of 5th group of the study in introduction and material and methods. How it could be related to scenario, as they first supplemented the birds with curcumin only and then after 14 days with both curcumin + imidacloprid. If they want to establish the protective effect of curcumin against toxicity, then must had withdrawn it from feed and then start using imidacloprid.

7.      The number of replicate are too less.

8.      The experimental design and group as explained in figure 1 not fully explained in words in the section of Material and Methods. E.g. Group 3: There is no written statement that imidacloprid was started after 14 days.

9.      The legends of graphs in figure 2 for groups are too close so it is suggested to use some alphabets (Group A, B, C…..) for groups and then those alphabets could be explained in descriptive legend of figure.

10.  The scale bar is 100 um for all, so needed to be corrected from spleen micrographs descriptive legends.

Comments on the Quality of English Language

Need extensive revision

Author Response

Comment 1: The research is very field oriented and applicable.

Response 1: Many thanks for your comment and efforts towards improving our manuscript.

Comment 2: The use of colon (:) in title is unnecessary.

Response 2: Done and corrected as recommended

Comment 3: Abstract: The first sentence (line 36 to 41) of abstract is too lengthy and not elaborative and need improvement to make the sense clearer.  

Response 3: Many thanks for your comment. It was done and the first sentence was rewritten again as recommended by the reviewer.

Comment 4: Abstract: the use “neonicotinoids” class of insecticide and then using the precise member of class name “imidacloprid” in same sentence can be confusing, so I will suggest using “imidacloprid” only.

Response 4: Many thanks for your helpful suggestion. Done and the corrected as recommended by the reviewer.

Comment 5: The article is lots of language issue, which need serious consideration.

Response 5: Thank you for your valuable comment, the manuscript has been checked for grammatical errors and the all corrections were done as recommended and highlighted in the manuscript. The manuscript was checked for language and grammar by MDPI. Assignment number: english-73492. The certificate for MDPI is uploaded.

Comment 6: The authors fail to establish the sense of inclusion of 5th group of the study in introduction and material and methods. How it could be related to scenario, as they first supplemented the birds with curcumin only and then after 14 days with both curcumin + imidacloprid. If they want to establish the protective effect of curcumin against toxicity, then must had withdrawn it from feed and then start using imidacloprid.

Response 6: The main aim of the study as mentioned at line 138 is to investigate the whether CUR supplementation either in the pro and concurrent or concurrent supplementation alone could mitigate the immuno-toxic effect of IMI significantly in broiler chickens. As CUR supplementation from the first day of experimental protocol could improve birds' immunity more significantly than if it is administered along with the IMI insecticide. So it is recommended to be used as additive from one day old chicks' diets.

Comment 7: The number of replicate are too less.

Response 7: Many thanks for this notable observation. We are apologizing for such unintended typo error. The number of used replicates were 3 per each experimental group. It was revised and corrected in the manuscript.

Comments 8: The experimental design and group as explained in figure 1 not fully explained in words in the section of Material and Methods. E.g. Group 3: There is no written statement that imidacloprid was started after 14 days.

Response 8: Many thanks for your thoughtful comment. The 3rd experimental group were described in section of material and methods line 162-163. 3rd (IMI): imidacloprid group, fed on control diet for 14 days then fed on a supplemented diet with a dose of IMI 50 mg/kg diet.

Comments 9: The legends of graphs in figure 2 for groups are too close so it is suggested to use some alphabets (Group A, B, C…..) for groups and then those alphabets could be explained in descriptive legend of figure.

Response 9: Many thanks for your comment. Done as recommended by reviewer. The legends of graphs in figure 2, 3, and 4 were modified as recommended. Group legends were corrected.

Comments 10: The scale bar is 100 um for all, so needed to be corrected from spleen micrographs descriptive legends.

Response 10: Many thanks for your comment. Done and corrected for all figure legends as recommended. For figure 5 scale bar was 100µm but for figure 6 and 7 the scale bar was 25µm and corrected.

Please address all correspondence concerning this manuscript to me at

[email protected]

Thank you for your consideration of this manuscript.

Sincerely, Ehsan H. Abu-Zeid
